# Extraction of Interface-Trap Densities of the Stacked Bonding Structure in 3D Integration Using High-Frequency Capacitance-Voltage Technique

**DOI:** 10.3390/mi13020262

**Published:** 2022-02-06

**Authors:** Man Li, Yufeng Guo, Jiafei Yao, Jun Zhang, Fanyu Liu, Weihua Tang

**Affiliations:** 1College of Electronic and Optical Engineering and College of Microelectronics, Nanjing University of Posts and Telecommunications, Nanjing 210023, China; qiqing0206@njupt.edu.cn (M.L.); zhangjun1991@njupt.edu.cn (J.Z.); whtang@njupt.edu.cn (W.T.); 2National and Local Joint Engineering Laboratory for RF Integration and Micro-Packaging Technologies, Nanjing University of Posts and Telecommunications, Nanjing 210023, China; 3Institute of Microelectronics, Chinese Academy of Sciences, Beijing 100029, China; liufanyu@ime.ac.cn

**Keywords:** high-frequency capacitance–voltage, interface-trap density, stacked bonding structure

## Abstract

An extraction method of the interface-trap densities (*D_it_*) of the stacked bonding structure in 3D integration using high-frequency capacitance–voltage technique is proposed. First, an accurate high-frequency capacitance–voltage model is derived. Next, by numerically solving the charge-balance equation and charge conservation equation, *D_it_* is extracted by fitting the measured and calculated capacitance–voltage curves based on the derived model. Subsequently, the accuracy of the derived model is verified by the agreements between the analytical results and TCAD simulation results. The average extraction error proves the precision and efficiency of the extraction method. Finally, the stacked bonding structure has been fabricated, and *D_it_* at the interface between silicon and insulator is extracted to diagnose and calibrate the fabrication processes.

## 1. Introduction

Three-dimensional (3D) integration has emerged as one possible approach to overcome the challenges of “More Moore” applications with the advantages of high-density integration, small form factor, high performance, low power consumption and multiple functionality of integrated circuits (ICs) [1,2]. It sequentially stacks multi-layer integrated circuit chips and then realizes electrical signal connection between multiple layers through monolithic inter-tier vias [3]. Wafer/chip stacked technology is one of the key technologies to realize 3D integration. Depending on whether the integration is performed at wafer or die level, there are three stacking options in 3D integration: wafer-to-wafer, chip-to-wafer and chip-to-chip [4,5]. At present, metal-to-metal direct bonding technology is an attractive option in the next generation of power devices and 3D IC technology, such as Au–Au, Cu–Cu or Al–Al [6,7,8,9]. Figure 1a shows the 3D view of a basic test structure of bonding technology in 3D integration, which can be applied to process the development of 3D integration prior to ICs’ fabrication and monitoring unit of fabricating 3D ICs. From top to bottom, it is divided into five layers: the silicon layer, the insulator layer, the metal layer, the insulator layer and the silicon layer. Note that the top and bottom silicon layers are both p-type and the two insulator layers have the same thickness.

The interface-trap density (*D_it_*) at the interface between silicon and insulator is one of the determinants of overall performance. Fabrication materials, processes and environment in 3D integration often produce different *D_it_*, which will affect the DC characteristics, AC performance, 1/*f* noise, crosstalk, etc. [10,11,12]. Therefore, it is important to extract *D_it_* of the 3D stacked bonding structure to improve performance, diagnosis and calibrate designs and fabrication processes, including bonding, annealing, oxidation, etc. The capacitance–voltage method has been widely used in characterizing cross-sectional characteristics. Conventionally, the high-frequency capacitance–voltage method has been frequently used to extract information about interface traps [13,14,15,16].

In this paper, an extraction method of *D_it_* of the stacked bonding structure in 3D integration is developed using a high-frequency capacitance–voltage technique. An accurate high-frequency capacitance–voltage model is derived, and *D_it_* is extracted by fitting the measured and calculated capacitance–voltage curves based on the derived model, which is verified by Sentaurus technology computer aided design (TCAD). The stacked bonding structure has been fabricated, and *D_it_* is extracted using the developed extraction method. Although the investigated structure presents complicated high-frequency capacitance–voltage characteristics with interface traps, the method is accurate and independent of the shape of high-frequency capacitance–voltage curves.

## 2. Theoretical Models

### 2.1. Accurate High-Frequency Capacitance–Voltage Model

As shown in Figure 1a, the stacked bonding structure can be equivalent to two back-to-back Metal–Insulator–Semiconductor (MIS) structures. Thus, the total high-frequency capacitance can be expressed as:(1)$$C_{total}^{HF}{}^{-1}=C_{HF1}{}^{-1}+C_{HF2}{}^{-1}$$
where *C_HF_*_1_ and *C_HF_*_2_ are the high-frequency capacitors of the upper and lower MIS structures. Since the interface-trapped charges cannot keep up with the change of the ac signal at high frequency, it does not contribute to the high-frequency capacitance. The high-frequency capacitance of MIS structure is a series connection of a semiconductor space-charge capacitance and an insulator capacitance. As Figure 1b shows, the total high-frequency capacitance can be rewritten as
(2)$$C_{total}^{HF}{}^{-1}=C_{s1}{}^{-1}+C_{s2}{}^{-1}+2C_{i}{}^{-1}$$
where *C_i_* is the capacitance per unit area of the insulator layer, which depends on the permittivity *ε_i_* and thickness *t_i_* of the insulator layer and follows the equation: $C_{i}=\varepsilon_{i}/t_{i}$. *C_s_*_1_ and *C_s_*_2_ are the differential capacitances of the upper and lower silicon layers.

When the state of silicon surface is in accumulation or depletion, *C_s_*_1_ and *C_s_*_2_ can be derived as follows [17]
(3)$$C_{s1,2}^{acc/dep}(V_{s1,2})=\left|\partial Q_{s1,2}/\partial V_{s1,2}\right|$$
where *V_s_*_1,2_ are the surface potentials of two silicon layers. *Q_s_*_1,2_ are the space-charge densities in the upper and lower silicon layer and can be given by [17]
(4)$$Q_{s1,2}(V_{s1,2})=-sign(V_{s1,2})\frac{2\varepsilon_{S}kT}{qL_{D1,2}}F\left(\frac{qV_{s1,2}}{kT},\frac{n_{p01,02}}{p_{p01,02}}\right)$$
where $F\left(a,b\right)=\sqrt{\left[\exp(-a)+a-1\right]+b\left[\exp(a)-a-1\right]}$, *k* is the Boltzmann constant, *T* is the absolute temperature, *q* is the electronic charge, *ε_s_* is the relative dielectric constant of the silicon layer. *L_D_*_1,2_ are the extrinsic Debye lengths in two silicon layers, and *n_p_*_01,02_ and *p_p_*_01,02_ are the equilibrium densities of electrons and holes in the upper and lower silicon layers, respectively.

For the state of silicon surface in strong inversion, *C_s_*_1_ and *C_s_*_2_ can be approximated as [17]
(5)$$C_{s1,2}^{stronginv}\approx \varepsilon_{s}/W_{dm1,2}$$

Here, *W_dm_*_1,2_ are the maximum widths of the depletion region in the upper and lower silicon layers, depending on the doping concentration *N_A_*.

As to the state of silicon surface in weak inversion or moderate inversion, an empirical function is used to achieve a continuous and smooth curve as shown in Figure 2, which is a small part of the high-frequency capacitance–voltage curve. We set
(6)$$C_{s1,2}^{weak/moderateinv}\left(V_{s1,2}\right)=\left(C_{s1,2}^{midgap}-C_{s1,2}^{stronginv}\right)f\left(V_{s1,2}\right)+C_{s1,2}^{stronginv}$$
in which $C_{s1,2}^{midgap}$ are the capacitances per unit area of two silicon layers with the Fermi level at midgap [17]. *f* (*V_s_*_1,2_) satisfies the following conditions
$$\begin{array}{l} f\left(V_{s1,2}=V_{s1,2}^{midgap}\right)=1\\ f\left(V_{s1,2}=V_{s1,2}^{strong\;inv}\right)=0\\ {\frac{df\left(V_{s1,2}\right)}{dV_{s1,2}}|}_{V_{s1,2}=V_{s1,2}^{strong\;inv}}=0 \end{array}$$

Thus, we can set the empirical function as $f\left(V_{s1,2}\right)=\frac{\sinh^{2}\left[\left(1-V_{s1,2}/V_{s1,2}^{strong\;inv}\right)/2\right]}{\sinh^{2}\left[\left(1-V_{s1,2}^{midgap}/V_{s1,2}^{strong\;inv}\right)/2\right]}$. $V_{s1,2}^{midgap}$ and $V_{s1,2}^{strong\;inv}$ are the surface potentials at midgap and the beginning of strong inversion in the upper and lower MIS structures, respectively. As shown in Figure 2, the analytical results under different structural parameters are in good agreement with TCAD simulated results, which validates the accuracy and usability of the high-frequency capacitance–voltage model.

### 2.2. D_it_ Extraction

As shown in Figure 1a, considering the influence of work-function difference and charges in the insulator layer, in addition to the interface traps, the net flat-band voltage *V_FB_* can be given by
(7)$$V_{FB}=\varphi_{ms}-\left(Q_{f}+Q_{t}+Q_{m}\right)/C_{i}$$
where *φ_ms_* is the work-function difference between the metal and silicon, *Q_f_*, *Q_t_* and *Q_m_* are the fixed charge density, trapped charge density and mobile ionic charge density in the insulator layer, respectively. Figure 3 shows the energy-band diagram of a staked bonding structure at flat band. Positive/negative insulator charges or lower/higher metal work function is equivalent to an added positive/negative bias.

Interface traps can be divided into two types: donor and acceptor. For simplicity, it is assumed that the distribution of trap levels is uniform [18]. Thus, the interface-trapped charge density *Q_it_* can be defined as follows
(8)$$Q_{it}\left(V_{s}\right)=q\left[\left|D_{it}\right|V_{s}-D_{it}\Delta V\right]$$
with
$$\left\{ \begin{array}{l} \Delta V=\frac{E_{g}}{2q}+\frac{kT}{q}\ln(\frac{p_{p0}}{n_{i}}),D_{it}>0\\ \Delta V=\frac{E_{g}}{2q}-\frac{kT}{q}\ln(\frac{p_{p0}}{n_{i}}),D_{it}<0 \end{array}\right.$$
where *V_s_* is the surface potential, *E_g_* denotes the band gap, *p_p0_* is the equilibrium density of holes, and *n_i_* denotes the intrinsic carrier concentration.

Note that, under high frequency case, it can be seen that different interface traps can keep up with the change of the applied voltage as shown in Figure 4. This results in the distortion of the high-frequency capacitance–voltage characteristics.

Based on the charge neutrality of the system, we can obtain
(9)$$Q_{M1}=-Q_{s1}\left(V_{s1}\right)+Q_{it1}\left(V_{s1}\right)$$
(10)$$Q_{M2}=-Q_{s2}\left(V_{s2}\right)-Q_{it2}\left(V_{s2}\right)$$
where *Q_it_*_1_ and *Q_it_*_2_ are the interface-trapped charge densities of the upper and lower MIS structures, respectively. *Q_M_*_1_ and *Q_M_*_2_ are the surface charge densities at the both sides of the metal layer.

Figure 5 shows three types of the *V_s_*_1_ and *V_s_*_2_ curves and corresponding energy-band diagrams with various interface trap densities. Two dashed lines are the dividing lines between accumulation, depletion and inversion at the silicon surface. The upper and lower silicon layers have the same doping concentration for facilitate analysis. When the investigated structure is biased with negative or positive voltages, the *V_s_*_1_ and *V_s_*_2_ curves intersect, namely, the two silicon layers have the same voltage drop *V_s_*^*^. With the increase in the interface-trap density, *V_s_*^*^ continuously increases, and the state of silicon surface changes gradually from accumulation, depletion, inversion in sequence, resulting in different energy-band diagrams as shown in Figure 5. Six cases may exist at two silicon surfaces for each type. Obviously, the voltage between the metal layer and the upper or lower silicon layer is *V_M_* – *V_g_* or *V_M_*, respectively. *V_M_* is the metal layer potential for the applied voltage *V_g_*. In the absence of work-function difference or insulator charges, the voltage, namely, *V_M_* − *V_g_* or *V_M_*, will appear partly across the insulator and partly across the silicon. Therefore, in conjunction with Figure 3, in addition to considering the influence of the substrate on the surface potential, the derivation of *Q_M_*_1_ or *Q_M_*_2_ is related to the applied voltage, the surface potential and the flat-band voltage
(11)$$Q_{M1}=C_{i}\left(V_{M}-V_{g}-V_{s1}-V_{sc1}+V_{FB1}\right)$$
(12)$$Q_{M2}=C_{i}\left(V_{M}-V_{s2}-V_{sc2}+V_{FB2}\right)$$
where *V_sc_*_1,2_ and *V_FB_*_1,2_ are the substrate calibration potentials and flat-band voltages of the upper and lower MIS structures, respectively. Among them, the ohmic-contact electrostatic potentials [19] are defined as the substrate calibration potentials to eliminate the effect of contact with the chuck or probe during testing or simulation, which can be determined by
$$V_{sc1,2}=-\frac{kT}{q}\ln\left(\frac{N_{A1,2}}{n_{i}}\right)$$
where *N_A_*_1,2_ are the doping concentrations of the upper and lower silicon layers.

Again, invoking the law of the conservation of electric charge, the amount of change in charge at the both sides of the metal layer is equal when a voltage is applied, giving
(13)$$Q_{M1}+Q_{M2}=Q_{M10}+Q_{M20}$$
where *Q_M_*_10_ and *Q_M_*_20_ are the surface charge densities at the both sides of the metal layer at zero-biased condition. Through Reference [20]
(14)$$Q_{M10}+Q_{M20}=C_{i}\left(-V_{s01}-V_{s02}-V_{sc1}-V_{sc2}+V_{FB1}+V_{FB2}\right)$$

Thus, Equation (14) may be recast as
(15)$$2V_{M}-V_{g}-V_{s1}-V_{s2}=-V_{s01}-V_{s02}$$
where *V_s01_* and *V_s_*_02_ are the surface potentials of two silicon layers when the applied voltage is set to 0 V, which can be calculated by Reference [20], respectively.

Figure 6 gives the flow of extracting interface-trap densities in the stacked bonding structure. First, given the initial value of *D_it_*_10_, *D_it_*_20_, *V_FB_*_10_ and *V_FB_*_20_, by combining Equations (7)–(10) and (15), we can compute the surface potentials of two silicon layers *V_s_*_1_ and *V_s_*_2_. Then, based on the derived accurate high-frequency capacitance–voltage model, substituting *V_s_*_1_ and *V_s_*_2_ into Equations (3)–(6) and (2) in sequence, the high-frequency capacitance as a function of the applied voltage can be calculated. Furthermore, using the nonlinear curve fitting function lsqcurvefit [21], the calculated high-frequency capacitance–voltage curve is fitted with the measured capacitance–voltage curve to extract the interface-trap densities of the upper and lower MIS structures *D_it_*_1_ and *D_it_*_2_.

## 3. Results and Discussion

In order to verify the precision and accuracy of the high-frequency capacitance–voltage model and *D_it_* extraction, the staked bonding structure is investigated through Sentaurus TCAD [22]. All structural parameters and their values of the stacked bonding structure used in simulation and experiment are listed in Table 1. The different mobility and recombination models are used in simulation, including the concentration dependent mobility model (CONMOB), the perpendicular electric field dependent mobility model (PRPMOB), the parallel electric field dependent mobility model (FLDMOB), the Shockley–Read–Hall recombination model with concentration dependent lifetimes (CONSRH) and Auger recombination model (AUGER).

Regardless of the interface traps, Figure 7 presents different high-frequency capacitance–voltage curves of the stacked bonding structure for various flat-band voltages. The thickness of the insulator layer is 0.2 μm, and the doping concentration of the upper and lower silicon layers are 9 × 10^14^ cm^−3^ and 1.5 × 10^14^ cm^−3^. The lines represent the analytical results of the proposed model, and the dots are the simulated results obtained using TCAD. The good agreement between the analytical and simulated results verifies the correction of the derived model. As Figure 7 shows, with various work-function differences or insulator charges, a parallel shift in the applied voltage bias direction occurs at the *C_HF_*_1_ or *C_HF_*_2_ curve. Therefore, the intersection of the *C_HF_*_1_ and *C_HF_*_2_ curves is changed, resulting in different high-frequency capacitance–voltage curves. At this time, the overlapping state of two silicon surfaces changes from strong inversion, moderate inversion, weak inversion, and depletion to accumulation. For strong inversion shown in Figure 7a, the high-frequency capacitance–voltage curve has a minimum platform. For moderate inversion shown in Figure 7b, the high-frequency capacitance–voltage curve has a minimum value. For weak inversion shown in Figure 7c, the high-frequency capacitance–voltage curve has a maximum value and a minimum value. For depletion shown in Figure 7d, the high-frequency capacitance–voltage curve has a maximum value. For accumulation shown in Figure 7e, the high-frequency capacitance–voltage curve has a maximum platform.

Figure 8a qualitatively shows the surface potentials of two silicon layers *V_s_*_1_ and *V_s_*_2_ as a function of the applied voltage with and without interface traps. Due to the fact that extra charges are needed to fill the traps, more charges or a larger applied voltage is required to obtain the same band bending or surface potential. As shown in Figure 8a, the *V_s_*_1_ and *V_s_*_2_ curves with interface traps are stretched out in the voltage direction compared with the curves without interface traps. Therefore, it also confirms Figure 8b, where the *C_HF_*_1_ and *C_HF_*_2_ curves are qualitatively drawn, with and without interface traps. As a result, the high-frequency capacitance–voltage curve of the staked bonding structure is changed with interface traps.

Figure 9 illustrates the analytical and simulated results of the high-frequency capacitance–voltage characteristics with various interface-trapped charge densities *D_it_*_1_ and *D_it_*_2_. The thickness of the insulator layer is 0.2 μm, and the doping concentrations of the upper and lower silicon layers are 9 × 10^14^ cm^−3^ and 1.5 × 10^14^ cm^−3^. The fixed charge density in the insulator layer is 10^10^ cm^−2^. The work function of the metal *φ_m_* is set to 4.65 eV. The tendency of the analytical results coincides well with the simulated results. As shown in Figure 9, the high-frequency capacitance–voltage curves saturate when the positive or negative bias is large enough. The saturated high-frequency capacitances under forward and reverse bias are associated with the maximal depletion region widths in the upper and lower silicon layer, respectively, as shown in Equation (5), which increases as the silicon doping concentration increases. Thus, as Figure 9 shows, the saturated capacitance under forward bias is significantly larger than the saturated capacitance under reverse bias. Although the saturated capacitance is independent of the interface traps, the critical applied voltage that reaches the saturated capacitance is greatly influenced by *D_it_*_1_ or *D_it_*_2_. The smaller *D_it_*_1_ or *D_it_*_2_, the smaller the critical positive/negative bias to reach the saturated capacitance. The high-frequency capacitance is severely sensitive to the interface traps when the applied voltage bias is small. As shown in Figure 9a, the capacitance increases with the acceptor-like interface traps and decreases with donor-like interface traps compared without interface traps. Furthermore, the capacitance increases/decreases gradually as the values of *D_it_*_1_ and *D_it_*_2_ increase. However, when $D_{it1}\neq D_{it2}$, the extreme-value no longer occurs at zero-biased condition. As shown in Figure 9b, with the increase/decrease of *D_it_*_2_, the applied voltage reaching an extreme value gradually shifts to the negative/positive bias direction.

Figure 10 shows the extraction results of *D_it_*_1,2_. The x-axis represents *D_it_*_1,2_ obtained by TCAD simulation, and the y-axis represents *D_it_*_1,2_ extracted by the proposed extraction method. The distance between the dots and diagonal line illustrates the accuracy of the extraction results. It can be seen from the figure that the dots are concentrated near the diagonal line, which verifies that this extraction method is accurate and independent of various high-frequency capacitance voltage curves.

## 4. Experiment

The stacked bonding structure was fabricated as shown in Figure 11. The substrate used P-type 4-inch Si-(100) wafer, whose thickness was 400 μm. After RCA cleaning and drying, a thermal oxide was grown to a thickness of about 200 nm in dry oxygen at 1100 °C. The dry oxygen time was 140 min. A 40 nm thick Cu was deposited, using a magnetron sputtering system. The base pressure during the deposition was in the range of 10^−3^–10^−2^ pa. Cu–Cu wafer bonding was carried out in a CB6L bonder. The chamber was evacuated to a base pressure of 6.3 × 10^−6^ Pa. The bonding temperature was 385 °C under a uniform bonding pressure of 10 kN and maintained for 45 min for the bonding process. Finally, the bonded wafer was annealed at 450 °C for 30 min in N_2_ ambient. Two batches of samples were fabricated to verify the capability of the proposed extraction method.

To evaluate the properties of the interfaces between silicon and insulator in a stacked bonding structure, high capacitance–voltage measurements were carried out using a Keysight B1505A analyzer (Keysight Technologies, Inc., Santa Rosa, CA, USA). A pressure-controlled probe was pierced on the upper silicon film, and the lower silicon film was in contact with the Cu chuck of the probe station. The measurement frequency was 1 MHz. The amplitude of measurement voltage was 0.2 V.

The measured and calculated high-frequency capacitance versus applied voltage for sample A or sample B is shown in Figure 12. The experimental and analytical results are normalized to have the same area. Through measurement and calculation, the thicknesses of the two insulator layers are both 0.2 μm. The doping concentrations of the top and bottom silicon layers of sample A are 9 × 10^14^ cm^−3^ and 1.5 × 10^14^ cm^−3^, respectively. The doping concentrations of the top and bottom silicon layers of sample B are 4 × 10^14^ cm^−3^ and 1.8 × 10^14^ cm^−3^, respectively. Under these structural parameters, the high-frequency capacitance versus applied voltage is calculated to fit the measured capacitance–voltage curve. As expected, it can be seen in Figure 12 that a good agreement between the experimental and analytical results is observed in general. The discrepancies in the analytical results are due to a uniform distribution of *D_it_*. Using the proposed extraction methodology, *D_it_*_1_ and *D_it_*_2_ of sample A are extracted as −7.2 × 10^11^ cm^−2^ eV^−1^ and −3.9 × 10^12^ cm^−2^ eV^−1^, and *D_it_*_1_ and *D_it_*_2_ of sample B are extracted as 5.7 × 10^11^ cm^−2^ eV^−1^ and −1.6 × 10^12^ cm^−2^ eV^−1^. It provides support for the diagnosis and calibration of the environments and designs in the fabrication processes such as deposition, bonding, annealing, etc.

## 5. Conclusions

In this paper, we have developed an accurate and reliable method to extract *D_it_* of the stacked bonding structure in 3D integration using high-frequency capacitance–voltage technique. An accurate high-frequency capacitance–voltage model has been proposed, and *D_it_* has been extracted based on the model. The results clearly demonstrate the accuracy of the derived model and the capability of the high-frequency capacitance–voltage technique performed in the stacked bonding structure to characterize the quality of the interfaces. The stacked bonding structure has been fabricated, and *D_it_* have been extracted from the measured high-frequency capacitance–voltage curve. The method has been shown to be an exact and non-destructive approach for diagnosing and calibrating 3D integration fabrication processes.

## Figures and Tables

**Figure 1 micromachines-13-00262-f001:**
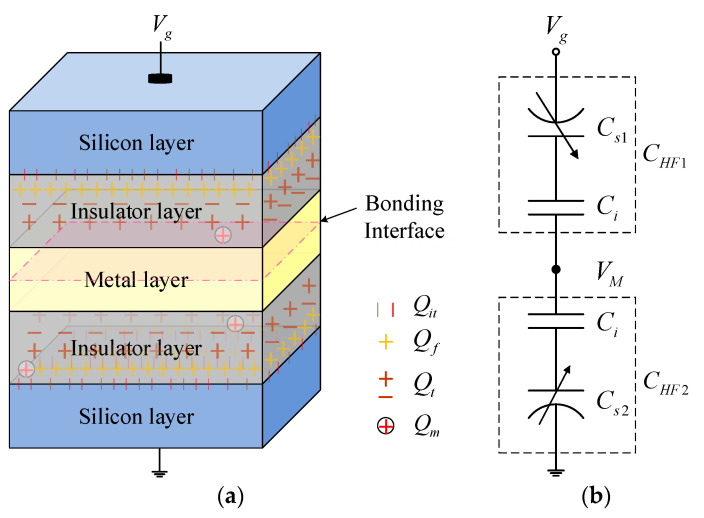
(**a**) Three-dimensional view and (**b**) high-frequency equivalent circuit model of the stacked bonding structure in 3D integration.

**Figure 2 micromachines-13-00262-f002:**
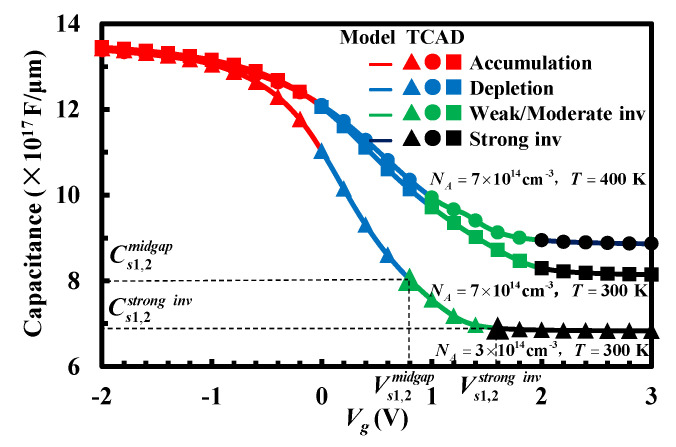
High-frequency capacitance–voltage curve of the upper or lower MIS structure in the stacked bonding structure.

**Figure 3 micromachines-13-00262-f003:**
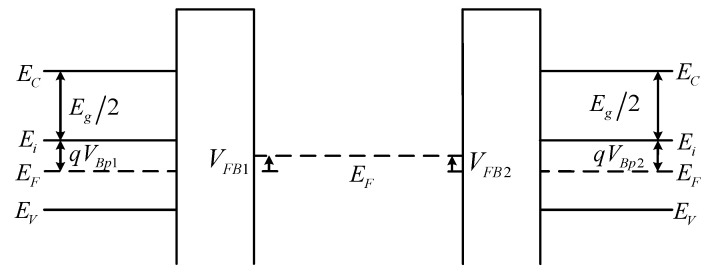
Energy-band diagram of a staked bonding structure at flat band.

**Figure 4 micromachines-13-00262-f004:**
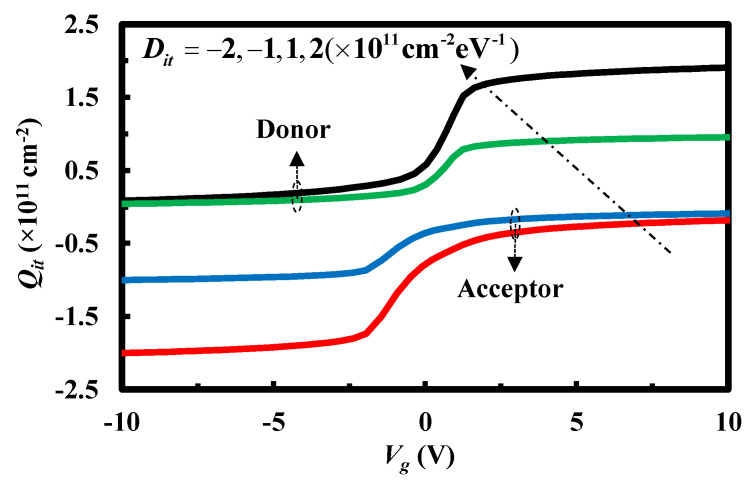
Interface-trapped charge density as a function of the applied voltage.

**Figure 5 micromachines-13-00262-f005:**
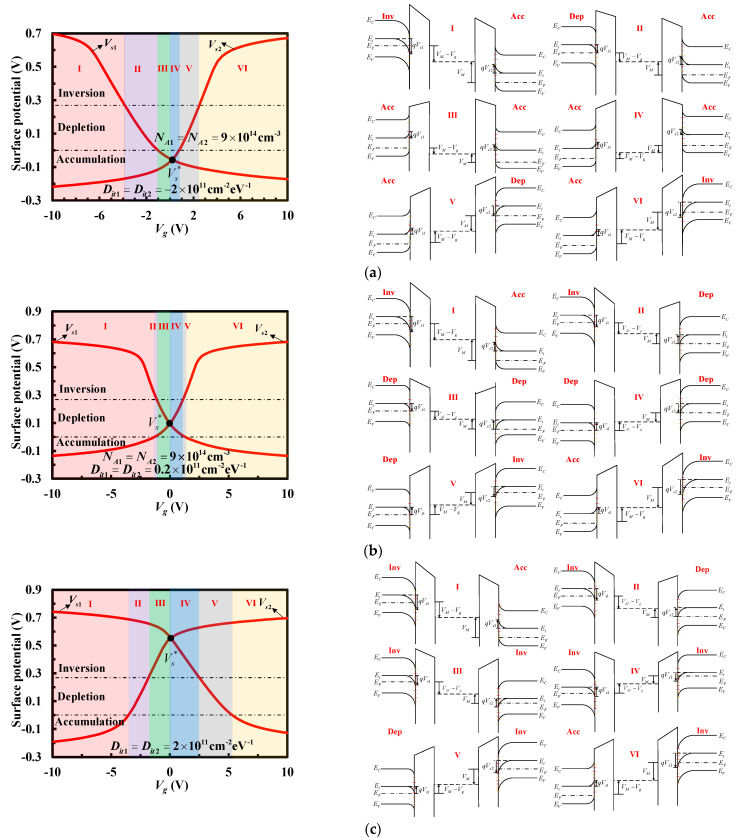
Surface potentials and energy-band diagrams: *V_s_*^*^ at (**a**) accumulation, (**b**) depletion, (**c**) inversion of the staked bonding structure.

**Figure 6 micromachines-13-00262-f006:**
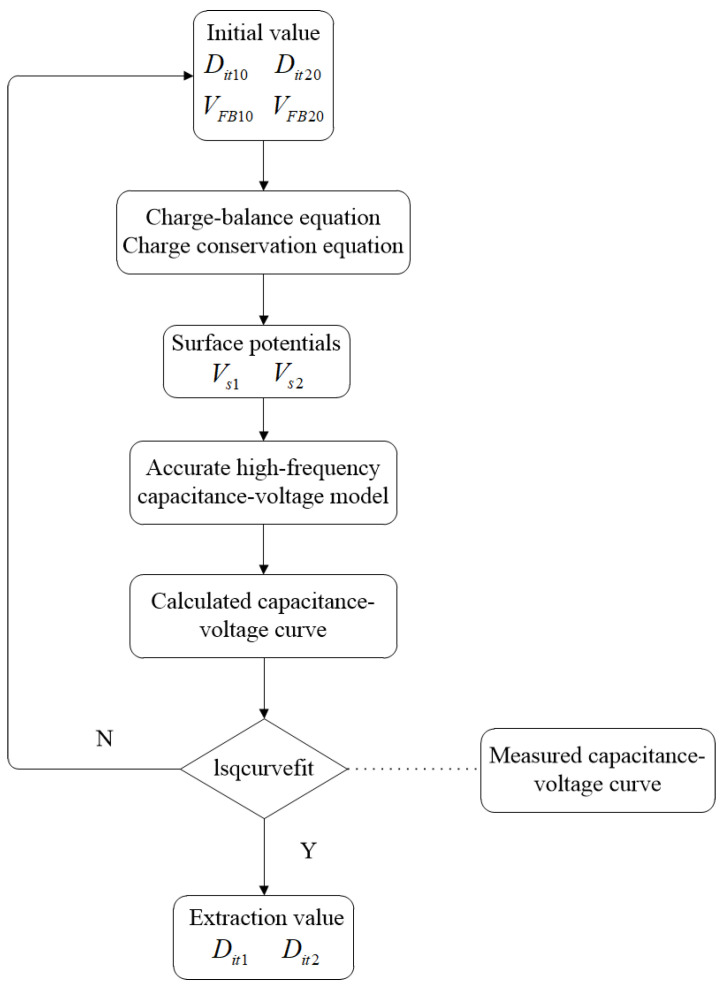
Flow chart of the extraction process of interface-trap densities in the stacked bonding structure.

**Figure 7 micromachines-13-00262-f007:**
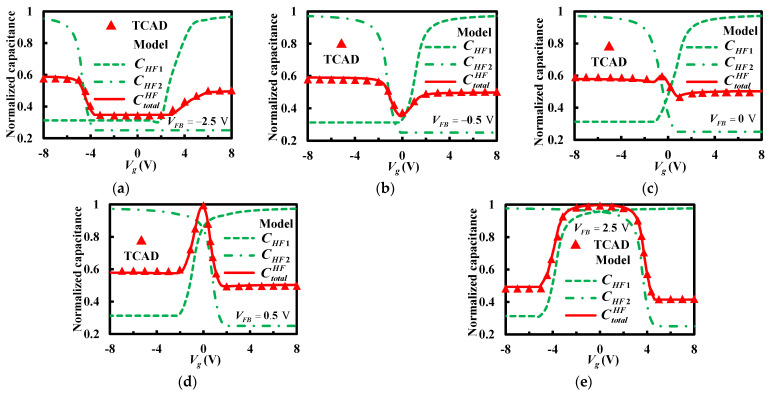
Different high-frequency capacitance–voltage curves of the staked bonding structure. (**a**) *V_FB_* = −2.5 V, (**b**) *V_FB_* = −0.5 V, (**c**) *V_FB_* = 0 V, (**d**) *V_FB_* = 0.5 V and (**e**) *V_FB_* = 2.5 V.

**Figure 8 micromachines-13-00262-f008:**
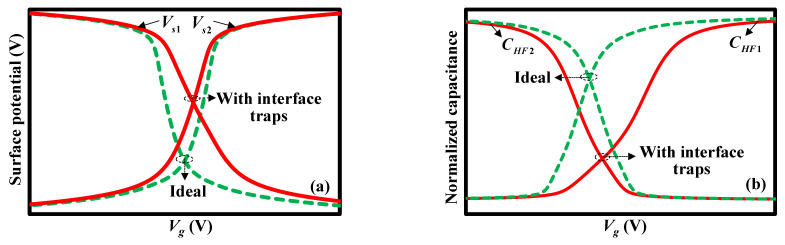
Influence of interface traps on (**a**) surface potentials and (**b**) high-frequency capacitance–voltage curves in the upper and lower MIS structures of the stacked bonding structure.

**Figure 9 micromachines-13-00262-f009:**
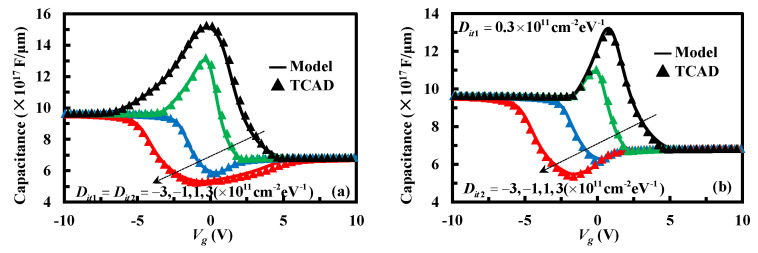
Under different interface-trap densities, high-frequency capacitance–voltage characteristics of the staked bonding structure: (**a**) $D_{it1}=D_{it2}$, (**b**) $D_{it1}\neq D_{it2}$.

**Figure 10 micromachines-13-00262-f010:**
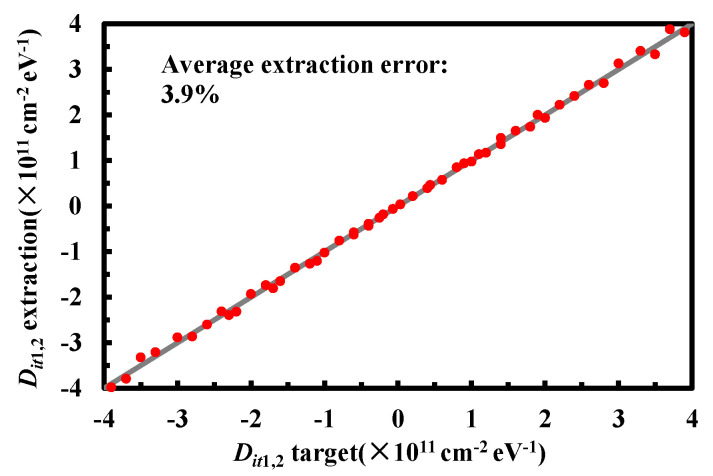
*D_it_*_1,2_ target and extraction using the proposed extraction method.

**Figure 11 micromachines-13-00262-f011:**
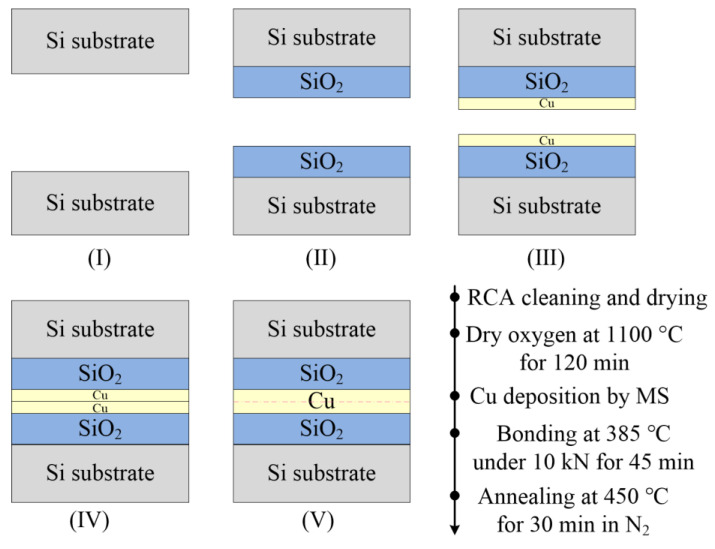
Process flow for fabrication of the stacked bonding structure.

**Figure 12 micromachines-13-00262-f012:**
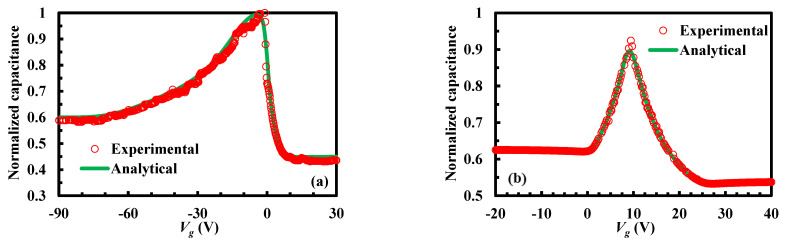
Measured and calculated high-frequency capacitance versus applied voltage of the stacked bonding structure: (**a**) sample A; (**b**) sample B.

**Table 1 micromachines-13-00262-t001:** Structural parameters and values of the stacked bonding structure.

Symbols	Quantity	Value
*t_i_*	thickness of the insulator layer	200 nm
*t_m_*	thickness of the metal layer	80 nm
*N_A_*	doping concentration of the silicon layer	1.5 × 10^14^–9 × 10^14^ cm^−3^
T	absolute temperature	300–400 K
*V_FB_*	flat-band voltage	−2.5–2.5 V
*φ_m_*	work function of the metal	4.65 eV
*Q_f_*	fixed charge density in the insulator layer	10^10^ cm^−2^
*D_it_*	interface-trapped charge density (simulation)	−4 × 10^11^–4 × 10^11^ cm^−2^ eV^−1^
*ε_s_*	permittivity of silicon	11.9
*ε_i_*	permittivity of insulator	3.9

## Data Availability

All authors declare that all data and materials generated or analyzed during this study are included in this article.

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
