# Peer review of "Extraction of Interface-Trap Densities of the Stacked Bonding Structure in 3D Integration Using High-Frequency Capacitance-Voltage Technique"

_micromachines, 2022, doi:10.3390/mi13020262_

Round 1

Reviewer 1 Report

No comments

Author Response

Our deepest gratitude goes to the reviewer for recognition of our manuscript. We will conduct more in-depth research in the future.

Reviewer 2 Report

I believe that the authors have adequately answered my comments on the previous manuscript version.

Minor comments:

  1. Check grammar in lines 58-59.
  2. Check the arrows direction in Fig. 6

Author Response

This manuscript is a resubmission of an earlier submission. The following is a list of the peer review reports and author responses from that submission.

Round 1

Reviewer 1 Report

  1. The major issue is the lack of novelty. I could see that the authors have done some prior work related to this which was published in conference. I do not see any significant addition to what is already known.
  2. The theoretical analysis presents a simplistic approach. the relevance of modeling which relates with stacked bonding structure is missing. It is a very simplistic analysis which is missing out on lot of details and is based on too many assumptions.

Reviewer 2 Report

This paper aims at proposing a method for extracting the interface traps in stacked bonding structure. The method consists in fitting the capacitance curve to a developed expression. Although the topic is interesting, I believe that there are some points that should be addressed by the authors before recommending it for publication:

  1. The model obtained in this work considers the series association of four capacitors related to the two insulation layers and the two semiconductor layers. When fabricating stacked integrated circuits, transistors are expected to be fabricated in the semiconductor active layer. How could this model be used when there are transistors in the layers? It seems that the model is applicable only at the passivation regions.
  2. I believe eq. (4) was not developed in the current work, so it must be referenced.
  3. (3) and (6) present two expressions for the capacitance, indicating that the former is used for the accumulation/depletion and the latter at inversion. However, it is not clear for me if both expressions are used for obtaining the curve of fig. 2. If so, how the transition between the two expressions is performed to assure continuity?
  4. In fig. 2, just one curve is shown to validate the model with TCAD results. However, eq. (6) uses an empirical function for the transition between strong and moderate inversions. Therefore, I believe that the model should be validated for different semiconductor doping concentrations and temperatures, since both parameters affect the depletion depth and, consequently, the semiconductor capacitance, to assure that the empirical function is valid for different conditions. Otherwise, the usability of the model would be limited.
  5. For the Dit extraction, the trap density was considered as uniform with either donor or acceptor trap type. In which conditions is this assumption valid? Generally, the interface trap profile in the MIS transistor presents a U-shape with maximum values close to the valence and conduction band, instead of a uniform profile. Besides, both acceptor and donor type could be present at different energy levels close to the valence and conduction band energies, instead of just one trap type. Therefore, I believe that the analysis should be extended considering also a non-uniform interface traps profile.
  6. In the unnumbered equation below eq. (12), a substrate calibration potential is defined. The potential in the equation seems to be the Fermi one.
  7. It is not clear the doping concentration used in fig. 5. Besides, how was the value of the dashed line in this figure defined? Was it calculated when the surface potential equals the Fermi potential?
  8. The simulator used, models considered and all the device characteristics should be presented in the text.
  9. No values are shown in the curves of fig. 7 and 8. Also, the trap density is not shown.

Reviewer 3 Report

The authors report a defect extract technique using high-frequency CV characteristics to qualify the wafer bonding. The idea seems to be interesting. However, some aspects of the study require further clarifications:

  1. I understand that the authors tried to simplify the device structure for simulation and fabrication. However, the device structure does not match the motivation of the study. The thickness of SiO2 is too thick for low-power devices. If the SiO2 layer were for metal-wiring isolation purposes, the Si/SiO2 interface should not be important. So, I suggest that the authors should use the proposed structure as a test unit that could be fabricated at the edge of the chip/wafer. Then, the wafer bonding quality could be evaluated using this device indirectly. In this case, the introduction should be modified to clarify this.
  2. Series resistance values were not considered in the equivalent circuit (Fig. 1). Series resistance plays an important role in the frequency dependence of the MOS device, especially low-doped semiconductors and high-frequency equivalent circuits. The extraction and consideration of the series resistance must be required.
  3. The device area is missed. Is the device not patterned? How big is the pressure-controlled probe area in comparison with the device area? If the probe area is big enough, the CV dispersion is not dependent on the contact configuration and reversely. The low resistance of the Si strongly affects the frequency-dependent CV characteristics through the contact configuration.
  4. It would be better to show how the trap density changes after bonding.
  5. More experimental data are suggested, such as different frequencies (Fig. 12).